# Extractive Spectrophotometric Determination and Theoretical Investigations of Two New Vanadium(V) Complexes

**DOI:** 10.3390/molecules28186723

**Published:** 2023-09-20

**Authors:** Kiril B. Gavazov, Petya V. Racheva, Antoaneta D. Saravanska, Galya K. Toncheva, Vasil B. Delchev

**Affiliations:** 1Department of Chemical Sciences, Faculty of Pharmacy, Medical University of Plovdiv, 120 Buxton Bros Str., 4004 Plovdiv, Bulgaria; 2Faculty of Chemistry, University of Plovdiv ‘Paisii Hilendarskii’, 24 Tsar Assen Str., 4000 Plovdiv, Bulgariavdelchev@uni-plovdiv.net (V.B.D.)

**Keywords:** vanadium, 6-hexyl-4-(2-thiazolylazo)resorcinol, tetrazolium, liquid–liquid extraction, spectrophotometric determination, HF and B3LYP calculations

## Abstract

Two new vanadium (V) complexes involving 6-hexyl-4-(2-thiazolylazo)resorcinol (HTAR) and tetrazolium cation were studied. The following commercially available tetrazolium salts were used as the cation source: tetrazolium red (2,3,5-triphenyltetrazol-2-ium;chloride, TTC) and neotetrazolium chloride (2-[4-[4-(3,5-diphenyltetrazol-2-ium-2-yl)phenyl]phenyl]-3,5-diphenyltetrazol-2-ium;dichloride, NTC). The cations (abbreviated as TT^+^ and NTC^+^) impart high hydrophobicity to the ternary complexes, allowing vanadium to be easily extracted and preconcentrated in one step. The complexes have different stoichiometry. The V(V)–HTAR–TTC complex dimerizes in the organic phase (chloroform) and can be represented by the formula [(TT^+^)[VO_2_(HTAR)]]_2_. The other complex is monomeric (NTC^+^)[VO_2_(HTAR)]. The cation has a +1 charge because one of the two chloride ions remains undissociated: NTC^+^ = (NT^2+^Cl^−^)^+^. The ground-state equilibrium geometries of the constituent cations and final complexes were optimized at the B3LYP and HF levels of theory. The dimer [(TT^+^)[VO_2_(HTAR)]]_2_ is more suitable for practical applications due to its better extraction characteristics and wider pH interval of formation and extraction. It was used for cheap and reliable extraction–spectrophotometric determination of V(V) traces in real samples. The absorption maximum, molar absorptivity coefficient, limit of detection, and linear working range were 549 nm, 5.2 × 10^4^ L mol^−1^ cm^−1^, 4.6 ng mL^−1^, and 0.015–2.0 μg mL^−1^, respectively.

## 1. Introduction

Vanadium can be classified as a dispersed trace element with an average content in the upper continental crust of 97 mg kg^−1^ [1]. It enters the biosphere through natural phenomena (mechanical and chemical rock weathering, volcanism, forest fires, and aeolian processes) and human activity. The main sources of anthropogenic vanadium are ore mining and processing, fossil fuel combustion, agricultural chemicalization, and the production of glass, ceramics, pigments, rubber, redox batteries, plastics, and sulfuric acid [2,3,4,5,6,7]. Recently, its anthropogenic enrichment factors (AEFs) have increased: vanadium ranks first among the trace elements in the atmosphere and fourth among the trace elements in the world’s rivers [2,5,8]. The problem is global [5] and should not be underestimated, as the most common anthropogenic vanadium form—V(V)—is almost as toxic as mercury, arsenic, lead, and cadmium [9]. This form resembles phosphorous (V) in chemical behavior [10] and can interfere with important biochemical processes by inhibiting the activity of key enzymes such as phosphatases and kinases [11,12].

Numerous methods based on coordination compounds have been used to determine V(V). Spectrophotometric methods are simple, cheap, and convenient [13,14,15]. They are often coupled with preconcentration techniques, which can greatly increase their range of application [14,15,16,17,18,19,20,21]. In a previous paper, we described the application of the V(V)—6-hexyl-4-(2-thiazolylazo)resorcinol (HTAR) anionic chelate [VO_2_(HTAR)]^−^ [22] for the liquid–liquid extraction (LLE)–spectrophotometric determination of hydroxyzine hydrochloride in pharmaceuticals [23]. In the present paper, we set three goals:To study the LLE of the V(V)–HTAR species in the presence of tetrazolium salt;To find the ground-state equilibrium geometries of the extracted species using quantum chemical calculations at the B3LYP and HF levels of theory;To develop a competitive LLE–spectrophotometric method for determining V(V) in real samples.

Tetrazolium salts (TSs) are an important class of compounds with versatile uses in chemistry, biology, and physics [24,25,26,27,28,29,30,31]. Two different TSs were used in this study: 2,3,5-triphenyltetrazol-2-ium;chloride (tetrazolium red, TTC) and 2-[4-[4-(3,5-diphenyltetrazol-2-ium-2-yl)phenyl]phenyl]-3,5-diphenyltetrazol-2-ium;dichloride (neotetrazolium chloride, NTC) (Figure 1). They have found applications both as leuco dyes [24] and cationic-type ion-association reagents [25]. The association ability of their cations is governed by various factors such as molecular weight, number of tetrazolium rings, and nature of substituents in these rings [25,32,33]. Here, we concentrate on their application as ion-association reagents with special emphasis on the structure and extractability of the resulting complexes.

## 2. Results and Discussion

### 2.1. Absorption Spectra

The spectra of the extracted species are shown in Figure 2. The absorbance of the two complexes (**1** and **2**) at their absorption maxima (*λ*_max_) is practically the same. However, there is a small difference in the position of these maxima: *λ*_max_ (1) = 549 nm and *λ*_max_ (2) = 556 nm. The spectral band of the V(V)–HTAR–TTC complex (1) is slightly broader, which may be due to the aggregation of the extracted species [34].

The absorbance of the blank samples (1′ and 2′) at the absorption maxima of the corresponding complexes is low. Although the concentrations of the reagents under the optimal extraction conditions (see below) are higher for the HTAR–TTC system, the absorbance of the blank is lower. This is a prerequisite for achieving better repeatability with this TS.

### 2.2. Effect of pH and the Amount of Buffer

The effect of pH was studied using a series of ammonium acetate buffer solutions with a known pH (Figure 3). The V(V)–HTAR–TTC complex is maximally extracted in a pH range of 3.9–5.0. The maximum extraction range of the V(V)–HTAR–NTC complex is shorter: 4.4–5.0. In this range, the absorbance of both complexes is the same.

To elucidate the effect of the amount of buffer, a series of experiments were carried out at pH 4.7 (one of the two pH values for which the buffering capacity is maximal [35]). It was found that there was no difference in absorbance in the presence of 1, 2, or 3 mL of the buffer solution. For reasons of economy, all subsequent experiments were performed with 1 mL of the buffer.

### 2.3. Effect of Extraction Time

The effect of extraction time is shown in Figure 4. Two minutes are needed to reach extraction equilibrium in the V(V)–HTAR–TTC system. Equilibrium in the other system is established a little slower (2.5 min). Since the shaking rate is a parameter that is difficult to control and depends on the experimenter, in order to avoid random and systematic errors, it is recommended that the extraction time be extended to 2.5 min and 3.0 min, respectively.

### 2.4. Effect of HTAR and TS Concentrations

The effect of the HTAR and TS (TTC or NTC) concentrations is shown in Figure 5 and Figure 6, respectively. Saturation is more easily reached in the V(V)–HTAR–NTC system. The optimal reagent concentrations for the two systems, among other optimized parameters, are shown in Table 1.

### 2.5. Stoichiometry, Formulas, and Equations

The small differences in the spectra (Figure 2) and optimal extraction conditions (Table 1) indicate possible differences in stoichiometry due to aggregation of the extracted species. Therefore, a variety of methods were used to elucidate the formulas of the complexes (Table 2), including those applicable to compounds of the type A_n_B_m_ (where *n* = *m* > 1) [36,37].

Figure 7a and Figure 8a present the results of determining the V(V):HTAR molar ratio in the ternary complexes via the mobile equilibrium method [36]. They show that the molar ratios in the two complexes are not the same: 2:2 (for the complex deriving from TTC) and 1:1 (for the complex deriving from NTC). This can be seen from the slopes (*n*) of the obtained straight lines, which match well with the correct value of *m*.

A similar conclusion is reached when studying the TS:V(V) molar ratio. When TS = TTC, the molar ratio is 2:2 (Figure 7b), and when TS = NTC, the molar ratio is 1:1 (Figure 8b).

An additional independent method [37] was used to confirm the stoichiometry in the TTC complex (Figure 9). As can be seen, a straight line is obtained for a complex of type A_2_B_2_.

It is known from the literature [39,42,43] that Job’s method [38] can be used to distinguish 1:1-complexes from A_n_B_n_ (*n* > 1) complexes (e.g., A_2_B_2_, A_3_B_3_, A_4_B_4_, etc.). The criterion for distinction is the presence or absence of concavities at the ends of the isomolar curve. Although such a distinction is not always reliable, it can be judged from Figure 10 that the NTC complex is most probably of the A_1_B_1_ type (no concavities), and the TTC complex is of the A_n_B_n_ type (*n* > 1). If this conclusion is compared with the results presented above obtained via the mobile equilibrium method [36] and the dilution method [37], it can be deduced that *n*_TTC_:*n*_V_ = 2:2.

The performed experiments give reason to assume that the complex obtained from TTC is an aggregate obtained in the organic phase via the dimerization of two 1:1:1 complexes represented by the formula (TT^+^)[VO_2_(HTAR)]. Such dimerization was reported for similar extraction systems containing V(V) and 5-methyl-4-(2-thiazolylazo)resorcinol [44,45].

The complex of NTC is monomeric (NTC^+^)[VO_2_(HTAR)]. The ditertazolium cation in it is most likely +1 charged: NTC^+^ = [(NT^2+^)Cl^−^]^+^. This agrees with Tôei’s concept of low charges (*z* = ±1) of the ion-associable ions [32,46].

Equations of complex formation and extraction, based on information for the state of V(V) [13] and HTAR (H_2_L) [47,48] at the optimum pH value, are as follows: 2 H_2_VO_4_^−^ _(aq)_ + 2 H_2_L _(aq)_ + 2 TT^+^
_(aq) =_ [(TT^+^)[VO_2_(L)]]_2 (org)_ + 4 H_2_O(1)
H_2_VO_4_^−^ _(aq)_ + H_2_L _(aq)_ + NTC^+^
_(aq) =_ (NTC^+^)[VO_2_(L)] _(org)_ + 2 H_2_O(2)

### 2.6. Extraction Characteristics

Three independent methods were used to calculate the conditional extraction constant (*K*_ex_) characterizing Equation (1): the mobile equilibrium method [36] (Figure 7b, straight line 2), the dilution method [37] (Figure 9), and the Likussar–Boltz method [44,45,49] (Figure 10a).

The mobile equilibrium method (Figure 8b), Likussar–Boltz method (Figure 10b), Holme–Langmyhr method [50], and Harvey–Manning method [51] were used to determine the extraction constant for the NTC system (Equation (2)).

The obtained values, along with the values of distribution ratios (*D*) and fractions extracted (*E*), are given in Table 3.

### 2.7. Ground-State Equilibrium Geometries of the Cations

The optimized ground-state equilibrium geometries of the cations (TT^+^ and NTC^+^) are shown in Figure 11a,b. The five atoms in each of the tetrazolium rings are coplanar. The lengths of the N1–N5 and N2–N3 bonds in TT^+^ are equivalent (1.308 Å) and slightly shorter than the other bonds in this ring. The bond lengths between the atoms in the NTC^+^ rings are also close in length: N(1)–N(5) = 1.313 Å, N(2)–N(3) = 1.313 Å, N(1)–N(2) = 1.353 Å, N(3)–C(4) = 1.352 Å, and N(5)–C(4) = 1.353 Å (left-hand ring), and N(24)–N(28) = 1.306 Å, N(25)–N(26) = 1.305 Å, N(24)–N(25) = 1.348 Å, N(26)–C(27) = 1.351 Å, and N(28)–C(27) = 1.349 Å (right-hand ring). The lack of significant differences is consistent with the conclusion that the internal dimensions “are largely insensitive to the local environment” [30,52]. As seen in Figure 11b, the chloride ion is located near the right-hand tetrazolium ring (closest to N24). It is pincered by two hydrogen atoms (H(64) and H(61)) of the substituent groups.

### 2.8. Ground-State Equilibrium Geometries of the Complexes

The next step was to correctly pair each of the two cationic structures (Figure 11a,b) with the anion [VO_2_(HTAR)]^−^ optimized in a previous paper [23] (Figure 11c). Two different NTC^+^–[VO_2_(HTAR)]^−^ structures were constructed and fully optimized at the HF/3–21G theoretical level (Figure 12a,b). Better stacking between the counterions was observed in Structure 1 (Figure 12a). Its energy is 15 kJ mol^−1^ lower than that of Structure 2.

The structure of [(TT^+^)[VO_2_(HTAR)]]_2_ was optimized in two steps. In the first step, the parent ions were assembled in two different ways, as shown in Figure 13a,b. In structure M1, the O(23) of the VO_2_ group is located near the tetrazolium ring. In the more stable structure M2, both oxygen atoms from the VO_2_ group are involved in additional interactions. Oxygen(24) is near the tetrazolium ring, and O(23) participates in a hydrogen bond C(50)–H(65)···O(23). This structure has 20 kJ mol^−1^ lower energy.

Figure 14 shows structures of dimers obtained in the second step—pairing of monomers. The dimeric structure D1 (a) is derived from two structures, M1, the dimeric structure D2 (b) is derived from two structures, M2, and the dimeric structure D3 (c), is derived from one structure, M1, and one M2.

The formation of D1 from two M1 structures is attended with a change of the Gibbs free energy of Δ*G*°_298_ = 20 kJ mol^−1^ and heat effect of Δ*H*°_298_ = –38 kJ mol^−1^. 

The structure D2 (Figure 14b) has 46 kJ mol^−1^ lower energy than the energy of D1. The change of Δ*G*°_298_ for the formation of the dimer from two M2 fragments is 18 kJ mol^−1,^ and the heat effect is Δ*H*°_298_ = −44 kJ mol^−1^.

The formation of the third dimer D3 is accompanied by Δ*G*°_298_ = 51 kJ mol^−1^ and Δ*H*°_298_ = −12 kJ mol^−1^. The dimer complex has 53 kJ mol^−1^ higher energy than D2 and only 7 kJ mol^−1^ than D1. In other words, the energy analysis of the three V(V)–HTAR–TT complexes led to the following stability series: D2 > D1 > D3.

### 2.9. Analytical Characteristics and Application

The relationships between the measured absorbance and the V(V) concentration in the aqueous phase were investigated under optimal conditions (see Table 1). The linear regression equations and some associated parameters related to the application of these systems for the determination of V(V) are included in Table 4. Advantages of the TTC–HTAR system are a wider linear range, a lower LOD, a lower cost of TTC (in comparison to NTC), and the tolerance of higher amounts of side ions such as Al(III), Ba(II), Br^−^, Ca(II), Cl^−^, Cr(VI), I^−^, Mo(VI), NO_3_^−^, and Re(VII) (Table 5). Advantages of the NTC–HTAR system are the lower reagents concentrations (Table 1) and higher tolerable levels of Zn(II), Cd(II), Hg(II), Mg(II), Pb(II), F^−^, and HPO_4_^2−^ (Table 5).

The HTAR–TTC system was used to determine V(V) in real samples such as vanadium-depleted catalysts from sulfuric acid production and pharmaceuticals. The results of the catalyst analysis are shown in Table 6. They are statistically indistinguishable from those obtained via an alternative spectrophotometric method [53]. The results of V(V) determination in spiked pharmaceutical samples are displayed in Table 7. The recoveries were in the range of 97.2–105%, with relative standard deviations from 2.0% to 4.6%.

### 2.10. Comparison with Other Spectrophotometric Methods

Table 8 compares the performance of several spectrophotometric procedures for determining vanadium. The proposed HTAR–TTC procedure can be described as simple, cheap, fast, sensitive, and reliable. It does not require any special or expensive equipment and is easier to implement than DLLME-SFOD [54], UA-CPE [17,55], or MA-CPE [53]. It is faster than many procedures requiring prolonged extraction (10–20 min) [16,21,56], centrifugation + cooling (10 min) [54], sonification at elevated temperature + centrifugation + cooling (17–28 min) [17,55], incubation at elevated temperature + cooling (55–110 min) [53,57], or two-fold extraction [21]. The sensitivity is higher than that of most liquid–liquid extraction (LLE) procedures [16,19,20,21,58,59], and the amount of organic solvent used is less than that of some of them [20,21,44,58]. Reagents are commercially available, and there is no need for tedious syntheses as described in [21,56,58,60]. Finally, the procedure can be considered reliable as the optimal intervals are wide enough and none of the thoroughly investigated parameters is problematic.

## 3. Materials and Methods

### 3.1. Reagents and Chemicals

Reagents from Merck (Schnelldorf, Germany), Fluka (Buchs, Switzerland), and Loba Feinchemie (Fischamend, Austria) were used without additional purification as aqueous solutions. The standard solution of V(V) was prepared from NH_4_VO_3_ (Merck, puriss. p. a.) at a concentration of 2.0 × 10^−4^ mol L^−1^. Solutions of TTC (Loba Feinchemie, p.a.) and NTC (Fluka, for microbiology) were stored in dark flasks; *c*_TTC_ = 3 × 10^−3^ mol L^−1^ and *c*_NTC_ = 2 × 10^−3^ mol L^−1^. The azo dye HTAR (Merck) was dissolved in the presence of KOH [22,57,62]; *c*_HTAR_ = 2 × 10^−3^ mol L^−1^. A series of buffer solutions (pH 3.3–7.4) were made by mixing 2 mol L^−1^ solutions of acetic acid and ammonia. Chloroform was purified by distillation and used repeatedly according to safety regulations. Distilled water was used in all experiments.

### 3.2. Instrumentation

An Ultrospec 3300 pro scanning spectrophotometer (Garforth, UK), equipped with 10 mm path-length quartz (or glass) cuvettes, was used during the work. The pH measurements were made with a WTW InoLab 7110 pH meter (Weilheim, Germany) with a glass electrode.

### 3.3. Samples

Pharmaceuticals were purchased from a local pharmacy: (i) Marimer inhalation (2.2% hypertonic seawater); (ii) Sterimar for nasal hygiene (a 100% natural, purified seawater-based nasal spray); and (iii) a solution of 0.9% NaCl for intravenous infusion. Aliquots of 5 mL of these solutions were subjected to the extraction procedure.

Samples of spent silica-supported catalysts (from the sulfuric acid production) were provided by KCM SA, Plovdiv (Bulgaria). They were ground in a mortar and prepared for analysis according to a known procedure [59]. Aliquots of 0.5 mL of the obtained solutions were used for the analysis.

### 3.4. Procedures for Optimization and Determination of Extraction Constants

The following solutions were mixed in a 125 mL separatory funnel: V(V), ammonium acetate buffer, HTAR, and TS (TTC or NTC). Water was added to a total volume of 10 mL. The volume of chloroform was 5 mL (in the optimization experiments) or 10 mL (in the determination of the extraction constant; Figure 7, Figure 8, Figure 9 and Figure 10). After pouring the organic solvent, the funnel was stoppered and shaken for a fixed time. Part of the resulting chloroform extract was filtered through filter paper and transferred to the cuvette. Absorbance was measured against a simultaneously prepared blank containing no V(V).

### 3.5. Procedure for Determination of Distribution Ratios and Fractions Extracted

The liquid–liquid distribution ratios for each of the systems were calculated after considering the absorbance in single extraction (*A*_1_) and triple extraction (*A*_3_) experiments: *D* = *A*_1_/(*A*_3_ − *A*_1_). The single extraction and the first step of the triple extraction were performed with 10 mL of chloroform under the optimal extraction conditions (Table 1). The organic layers were transferred into 25 mL volumetric flasks. The flask for the single extraction was brought to the mark with chloroform. The second and third steps of the triple extraction were carried out by adding 7 mL of chloroform to the aqueous phase of the previous step. The organic extracts were combined in the second 25 mL volumetric flask, and chloroform was added to the mark. Absorbances were measured against corresponding blanks.

The fractions extracted were calculated using the equation %*E* = 100 × *D*/(*D* + 1).

### 3.6. Procedure for Determination of Vanadium(V) with HTAR and TTC

An aliquot of the analyzed solution (containing 0.015–2.0 μg mL^−1^ of V) was transferred in a 125 mL separatory funnel. Solutions of the buffer (1 mL, pH 4.7), HTAR (0.4 mL, 2 × 10^−3^ mol L^−1^), and TTC (0.8 mL, 3 × 10^−3^ mol L^−1^) were added, and the sample was diluted with water until a total volume of 10 mL. Then, 5 mL of chloroform was added, and the mixture was shaken for 2.5 min. After the phase separation, a portion of the chloroform extract was poured into the cuvette, and the absorbance was measured at 549 nm against a blank. The unknown V(V) concentration was calculated from a calibration plot prepared using the same procedure.

### 3.7. Procedure for Determination of Vanadium(V) with HTAR and NTC

An aliquot of the analyzed solution (containing 0.023–1.1 μg mL^−1^ of V) was transferred in a 125 mL separatory funnel. Solutions of the buffer (1 mL, pH 4.7), HTAR (0.2 mL, 2 × 10^−3^ mol L^−1^), and NTC (0.7 mL, 2 × 10^−3^ mol L^−1^) were added, and the sample was diluted with water until a total volume of 10 mL. Then, 5 mL of chloroform was added, and the mixture was shaken for 3.0 min. After the phase separation, a portion of the chloroform extract was poured into the cuvette, and the absorbance was measured at 556 nm against a blank. The unknown V(V) concentration was calculated from a calibration plot prepared using the same procedure.

### 3.8. Theoretical

The ground-state equilibrium geometries of the cations, TT^+^ and NTC^+^, were optimized at the B3LYP/6-311++G** and B3LYP/6-31+G* levels of theory, respectively. Each of these cations was then paired in different ways with the anionic complex [VO_2_(HTAR)]^−^ studied in a previous work [23] as described above. 

The NTC^+^ and [VO_2_(HTAR)]^−^ counterions were assembled in two ways and optimized at the HF/3-21G level of theory. Furthermore, the thermodynamic characteristics of the obtained structures were calculated and compared.

The TT^+^ and [VO_2_(HTAR)]^−^ counterions were also assembled in two different ways and optimized at the HF/3-21G level of theory to obtain monomeric structures (M1 and M2) of the type (TT^+^)[VO_2_(HTAR)]. These structures were combined into three dimers (M1–M1, M2–M2, and M1–M2). Optimization was performed at the HF/3-21G level of theory, and the stability of the obtained species was evaluated.

All calculations were performed for the gas phase using the GAUSSIAN 16 program package [63]. The ChemCraft program (https://chemcraftprog.com (accessed on 13 September 2023)) was used for the visualization of the computed results [64].

## 4. Conclusions

The complex formation of two new V(V) complexes has been studied. The differences between the extracted species have been highlighted. The behavior of the ditetrazolium salt NTC is interesting because of its participation in the final complex as a singly charged cation [(NT)^2+^Cl^−^]^+^. The TT^+^ complex, on the other hand, is notable for its dimeric structure, which gives it excellent extractability and good analytical characteristics. It has been applied to the analysis of real samples—pharmaceuticals and silica-supported catalysts.

## Figures and Tables

**Figure 1 molecules-28-06723-f001:**
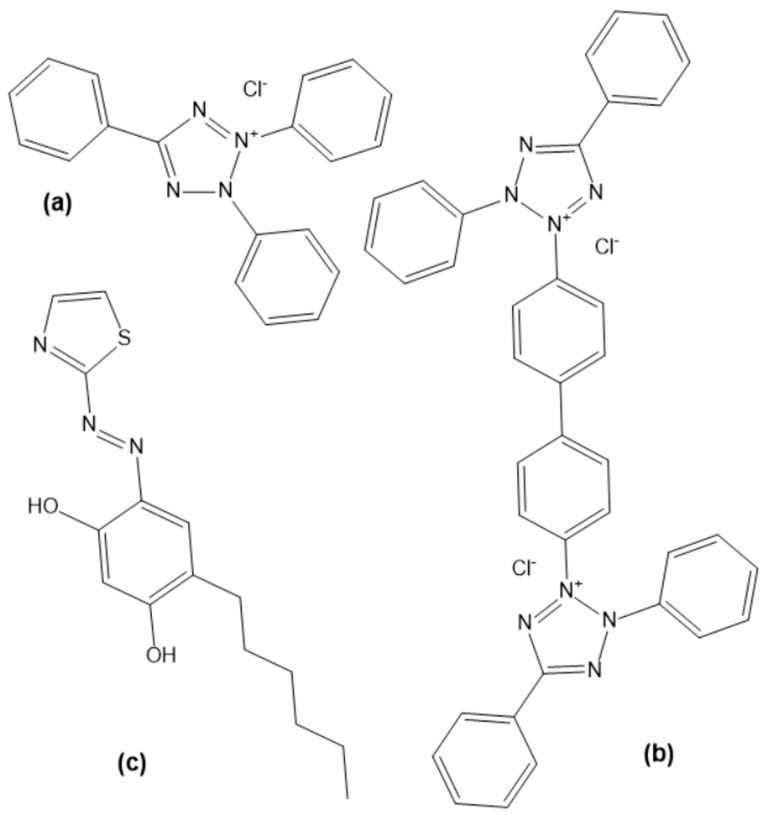
Structural formulae of the reagents TTC (**a**), NTC (**b**), and HTAR (**c**).

**Figure 2 molecules-28-06723-f002:**
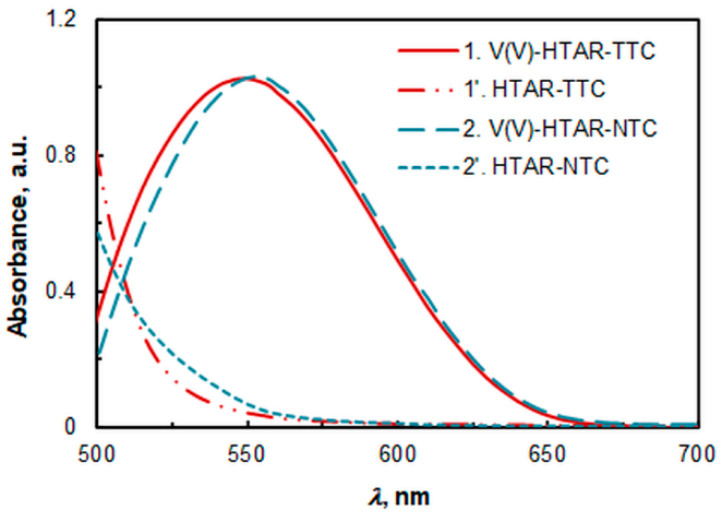
Absorption spectra of the V(V)-HTAR-TTC complex (1) and V(V)-HTAR-NTC complex (2) against corresponding blanks. The spectra of the blanks (1′ and 2′) are recorded against chloroform. *c*_V(V)_ = 2 × 10^−5^ mol L^−1^, *c*_HTAR_ = 8 × 10^−5^ mol L^−1^ (1, 1′) or 4 × 10^−5^ mol L^−1^ (2, 2′), *c*_TTC_ =2.4 × 10^−4^ mol L^−1^, *c*_NTC_ =1.4 × 10^−4^ mol L^−1^, and pH 5.0 (ammonium acetate buffer).

**Figure 3 molecules-28-06723-f003:**
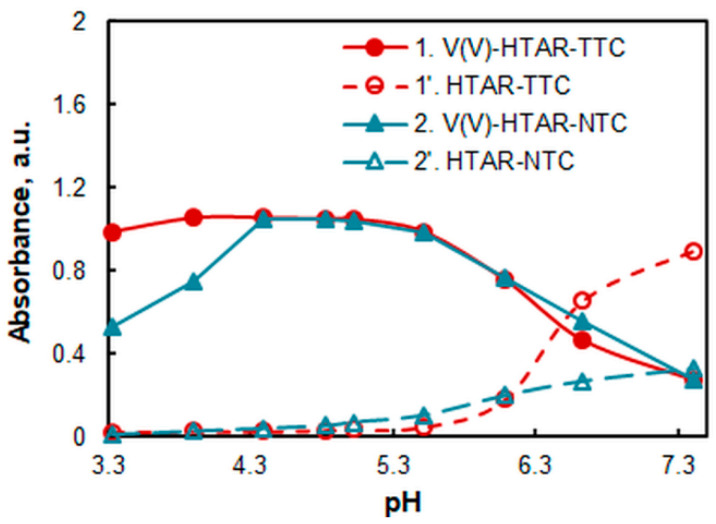
Absorbance of the complexes (1, 2) and blanks (1′, 2′) in chloroform vs. pH of aqueous phase. *c*_V(V)_ = 2 × 10^−5^ mol L^−1^, *c*_HTAR_ = 8 × 10^−5^ mol L^−1^ (1, 1′) or 4 × 10^−5^ mol L^−1^ (2, 2′), *c*_TTC_ = 2.4 × 10^−4^ mol L^−1^, *c*_NTC_ = 1.4 × 10^−4^ mol L^−1^, *λ* = 549 nm (1, 1′), or 556 nm (2, 2′).

**Figure 4 molecules-28-06723-f004:**
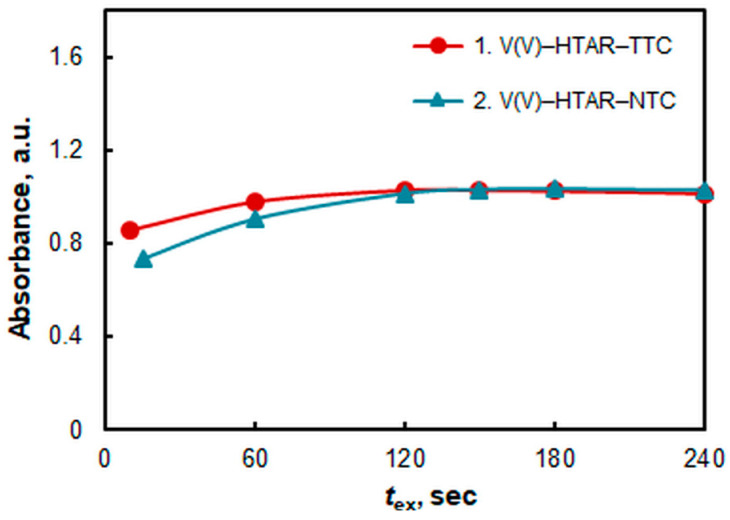
The effect of extraction time (*t*_ex_). *c*_V(V)_ = 2 × 10^−5^ mol L^−1^, *c*_HTAR_ = 8 × 10^−5^ mol L^−1^ (1) or 4 × 10^−5^ mol L^−1^ (2), *c*_TTC_ =2.4 × 10^−4^ mol L^−1^, *c*_NTC_ =1.4 × 10^−4^ mol L^−1^, and pH = 4.7.

**Figure 5 molecules-28-06723-f005:**
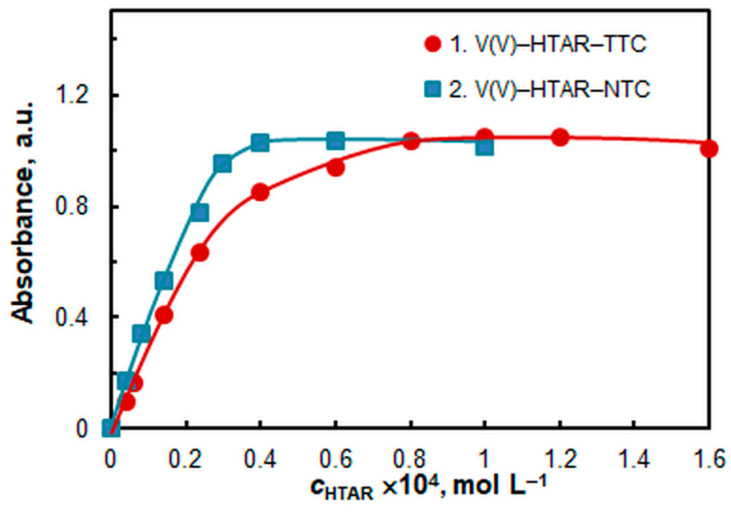
The effect of HTAR concentration. *c*_V(V)_ = 2 × 10^−5^ mol L^−1^, *c*_TTC_ =6 × 10^−4^ mol L^−1^, *c*_NTC_ =4 × 10^−4^ mol L^−1^, and pH = 5.0.

**Figure 6 molecules-28-06723-f006:**
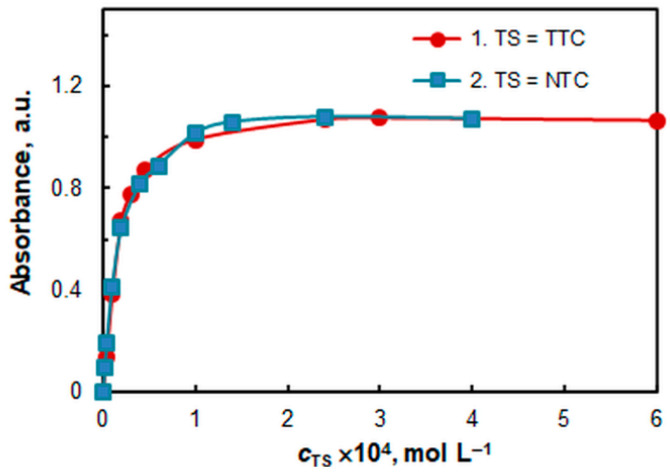
The effect of TS concentration. TS = TTC (1), TS = NTC (2). *c*_V(V)_ = 2 × 10^−5^ mol L^−1^, *c*_HTAR_ = 1 × 10^−4^ mol L^−1^ (1) or 4 × 10^−5^ mol L^−1^ (2), and pH = 5.0.

**Figure 7 molecules-28-06723-f007:**
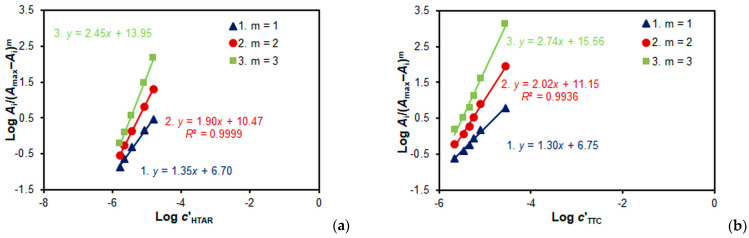
Determination of the HTAR:V(V) (**a**) and TTC:V(V) molar ratios in the V(V)–HTAR–TTC complex via the mobile equilibrium method. *c*_V(V)_ = 2 × 10^−5^ mol L^−1^*c*_TTC_ = 3 × 10^−4^ mol L^−1^ (**a**), and *c*_HTAR_ = 8 × 10^−5^ mol L^−1^ (**b**).

**Figure 8 molecules-28-06723-f008:**
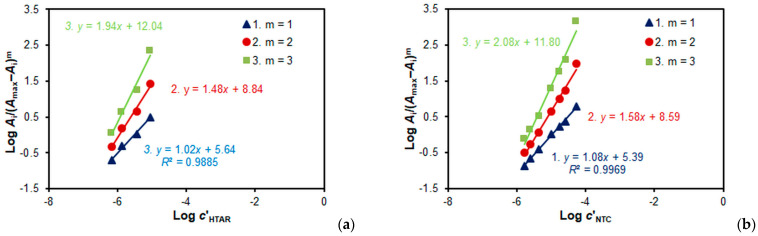
Determination of the HTAR:V(V) (**a**) and NTC:V(V) molar ratios in the V(V)–HTAR–NTC complex via the mobile equilibrium method. *c*_V(V)_ = 2 × 10^−5^ mol L^−1^, *c*_NTC_ = 4 × 10^−4^ mol L^−1^ (**a**), and *c*_HTAR_ = 4 × 10^−5^ mol L^−1^ (**b**).

**Figure 9 molecules-28-06723-f009:**
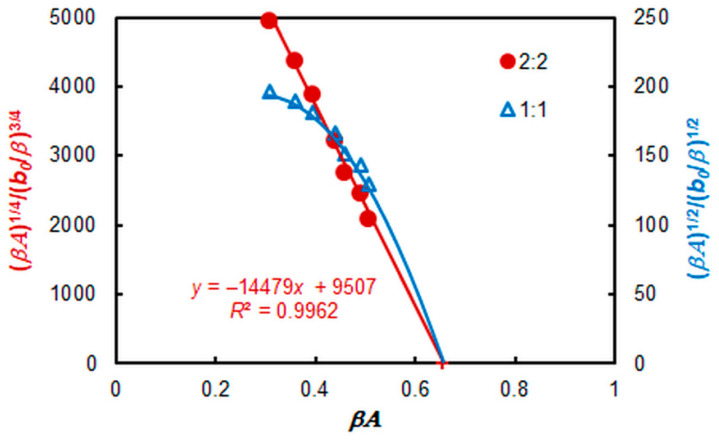
Determination of the TTC:V(V) molar ratio in the V(V)–HTAR–TTC complex via the dilution method. *c*_TTC_ = *c*_V(V)_, *b*_0_ = 3 × 10^−5^ mol L^−1^, *c*_HTAR_ = 8 × 10^−5^ mol L^−1^, and pH = 4.7.

**Figure 10 molecules-28-06723-f010:**
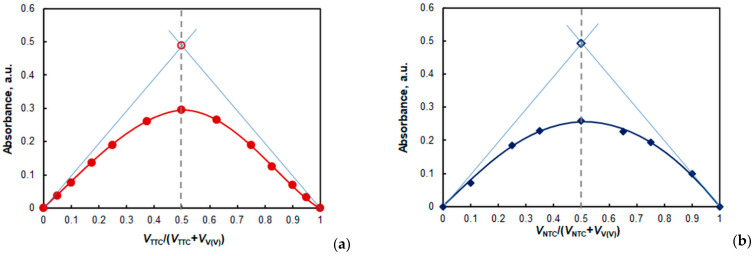
Job’s method of continuous variations and Likussar–Boltz approach for the determination of *K*_ex_ in the V(V)—HTAR—TTC system (**a**) and V(V)—HTAR—NTC system (**b**). *k* = *c*_V(V)_ + *c*_TS_ = 4 × 10^−5^ mol L^−1^, *c*_HTAR_ = 8 × 10^−5^ mol L^−1^ (**a**) or 4 × 10^−5^ mol L^−1^ (**b**), and pH = 4.7.

**Figure 11 molecules-28-06723-f011:**
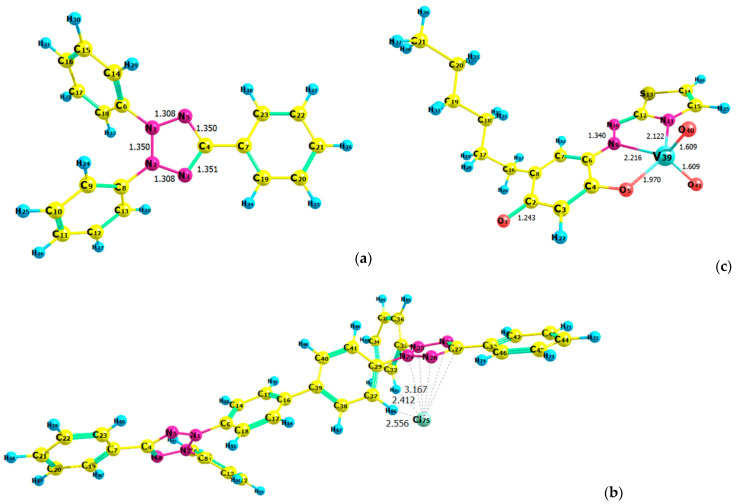
Optimized ground-state structures of the ions: TT^+^ (B3LYP/6-311++G**) (**a**), NTC^+^ (B3LYP/6-31+G*) (**b**), and [VO_2_(HTAR)]^−^ (B3LYP/6-311++G**) (**c**).

**Figure 12 molecules-28-06723-f012:**
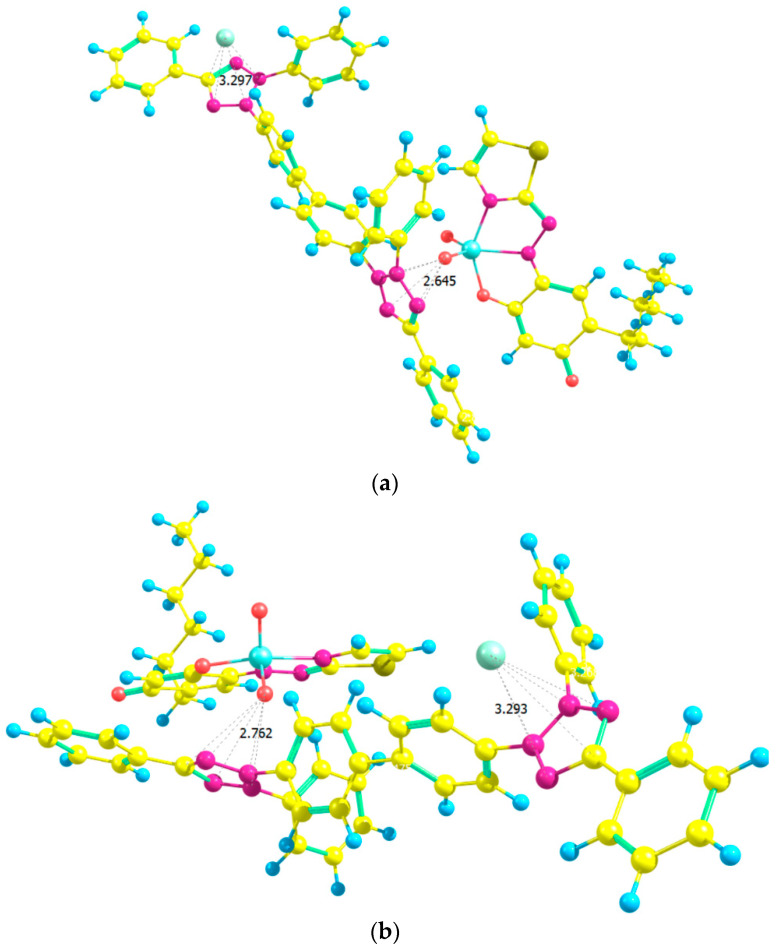
Optimized ground-state structures of the (NTC^+^)[VO_2_(HTAR)]^−^ ion-association complex at the HF/3-21G theoretical level: (**a**) Structure 1, *E*_1_ = −4706.2401 a.u., *G*_1_ = −4705.3824 a.u., *H*_1_ = −4705.2128 a.u.; (**b**) Structure 2, *E*_2_ = −4706.2345 a.u., *G*_2_ = −4705.3779 a.u., *H*_2_ = −4705.2073 a.u.

**Figure 13 molecules-28-06723-f013:**
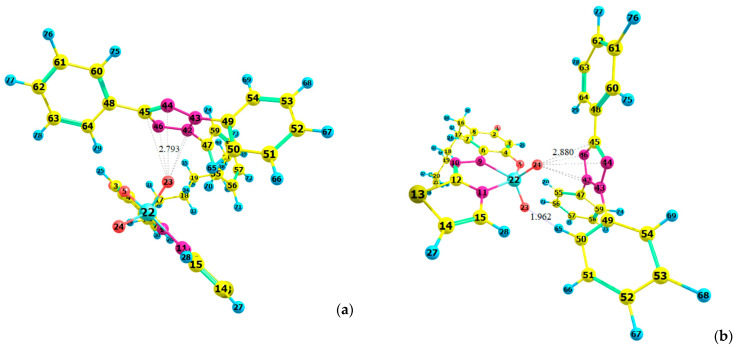
Optimized ground-state structures of the (TT^+^)[VO_2_(HTAR)] monomer at the HF/3-21G theoretical level: (**a**) Structure M1, *E*_1_ = −3309.4733 a.u., *G*_1_ = −3308.8904 a.u., *H*_1_ = −3308.7693 a.u.; (**b**) Structure M2, *E*_2_ = −3309.4656 a.u., *G*_2_ = −3308.8823 a.u., *H*_2_ = −3308.7619 a.u.

**Figure 14 molecules-28-06723-f014:**
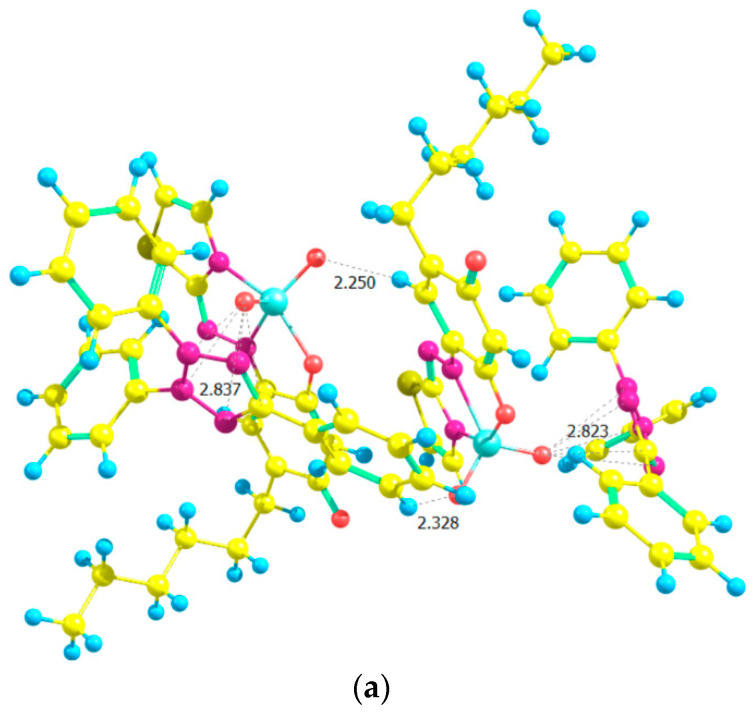
Optimized ground-state structures of the dimer [(TT^+^)[VO_2_(HTAR)]]_2_ at the HF/3-21G theoretical level: (**a**) Structure D1, *E*_1_ = −6618.9481 a.u., *G*_1_ = −6617.7571 a.u., *H*_1_ = −6617.5383 a.u.; (**b**) Structure D2, *E*_2_ = −6618.9658 a.u., *G*_2_ = −6617.7737 a.u., *H*_2_ = −6617.5555 a.u.; (**c**) Structure D3, *E*_3_ = −6618.9454 a.u., *G*_3_ = −6617.7531 a.u., *H*_3_ = −6617.5359 a.u.

**Table 1 molecules-28-06723-t001:** Optimum conditions ^a^.

Extraction System	λ_max_, nm	pH ^b^	*c*_HTAR_, mol L^−1^	*c*_TS_, mol L^−1^	Extraction Time, min
V(V)–HTAR–TTC	549	4.7	8 × 10^−5^	2.4 × 10^−4^	2.5
V(V)–HTAR–NTC	556	4.7	4 × 10^−5^	1.4 × 10^−4^	3.0

^a^ Optimization studies conducted at room temperature (22 ± 1 °C), *V*_aq_ = 10 mL, and *V*_chloroform_ = 5 mL. ^b^ Ammonium acetate buffer (1 mL).

**Table 2 molecules-28-06723-t002:** Molar ratios in the ternary V(V)–HTAR–TS complexes obtained via different methods.

Molar Ratio	Mobile Equilibrium Method [36] ^a^	Dilution Method [37] ^a^	Job’s Method [38,39] ^b^	Asmus’ Method [40] ^c^	Bent–French Method [41] ^c^
HTAR:V (TS = TTC)	2:2	–	–	1:1	1:1
HTAR:V (TS = NTC)	1:1	–	–	1:1	1:1
TTC:V	2:2	2:2	*n*:*n* (*n* > 1)	1:1	1:1
NTC:V	1:1	–		1:1	1:1

^a^ Methods applicable to complexes of the type A_2_B_2_. ^b^ A method capable of distinguishing A_1_B_1_ complexes from A_n_B_n_ (*n* > 1) complexes. ^c^ Other methods with more limited capabilities.

**Table 3 molecules-28-06723-t003:** Calculated values of conditional extraction constants (*K*_ex_), distribution ratios (*D*), and fractions extracted (%*E*).

Extraction System	Log *K*_ex_					Log *D*	%*E*
	Mobile Equilibrium Method [36]	Dilution Method [37]	Likussar–Boltz Method [49]	Holme–Langmyhr Method [50]	Harvey–Manning Method [51]		
V(V)—HTAR—TTC	15.2 ± 0.4 ^a^	15.8 ± 0.1 ^c^	15.2 ± 0.1 ^d^ 15.1 ± 0.1 ^e^	–	–	1.53 ± 0.08 ^h^	97.1 ± 0.5 ^h^
V(V)—HTAR—NTC	5.0 ± 0.1 ^b^	–	5.1 ± 0.1 ^f^	5.0 ± 0.1 ^g^	5.0 ± 0.1 ^h^	0.94 ± 0.14 ^i^	90 ± 3 ^i^

^a^ Figure 7b; ^b^ Figure 8b; ^c^ Figure 9; ^d^ Figure 10a, *k* = 4 × 10^−5^; ^e^
*k* = 1 × 10^−4^; ^f^ Figure 10b; ^g^ *N* = 7; ^h^
*N* = 3; ^i^ *N* = 4.

**Table 4 molecules-28-06723-t004:** Characteristics concerning the application of the two extraction–chromogenic systems for determining V(V).

Characteristics	HTAR–TTC System	HTAR–NTC System
Linear regression equation *y* = *ax* + *b*	*y* = 1.020*x* + 0.0003	*y* = 1.018*x* − 0.005
Correlation coefficient	0.9990 (*N* = 11)	0.9996 (*N* = 8)
Standard deviations of the slope (*a*) and *y*-intercept (*b*)	0.015 and 0.017	0.012 and 0.008
Linear range, µg mL^−1^	0.015−2.0	0.023−1.1
Molar absorptivity coefficient (*ε*), L mol^−1^ cm^−1^	5.2 × 10^4^	5.2 × 10^4^
Sandell’s sensitivity, ng cm^−2^	0.98	0.98
Limit of detection (LOD) ^a^, ng mL^−1^	4.6	6.8
Limit of quantitation (LOQ) ^b^, ng mL^−1^	15	23

^a^ Three times the standard deviation (SD) of the blank divided by the slope. ^b^ Ten times the SD of the blank divided by the slope.

**Table 5 molecules-28-06723-t005:** Effect of foreign ions on determining 5 μg vanadium(V).

Foreign Ion (FI) Added	Added Salt Formula	HTAR–TTC System			HTAR–NTC System		
		FI:V(V) Mass Ratio	Amount of V(V) Found, μg	R, %	FI:V(V) Mass Ratio	Amount of V(V) Found, μg	R, %
Al(III)	Al_2_(SO_4_)_3_·7H_2_O	1000	5.22	104	500	4.84	96.7
Ba(II)	Ba(NO_3_)_2_	2000 ^a^	4.84	96.8	100 1000	5.01 4.42	100 88.5
Br^−^	NaBr	2000 ^a^	4.93	98.6	250	4.93	98.5
Ca(II)	Ca(NO_3_)_2_·4H_2_O	2000 ^a^	5.01	100	500	4.91	98.2
Cd(II)	3CdSO_4_·8H_2_O	5 500	5.13 5.89	103 118	50 500	5.24 4.97	105 99.4
Cl^−^	NaCl	10,000 ^a^	5.02	100	250 2000	4.81 4.58	96.3 91.6
Co(II)	CoSO_4_·7H_2_O	0.5	5.16	103	0.5	5.24	105
Cr(III)	Cr_2_(SO_4_)_3_	4	4.83	96.7	5	5.19	104
Cr(VI)	KCrO_4_	500	4.87	97.4	250	4.94	98.7
Cu(II)	CuSO_4_·5H_2_O	0.5	4.84	96.7	1	5.10	102
F^−^	NaF	500 1000	4.84 4.54	96.7 90.7	2000 5000	5.02 4.40	100 88.0
Fe(III)	Fe_2_(SO_4_)_3_	0.5	5.28	106	0.5 10 ^b^ 50 ^b^	5.42 5.10 5.40	108 102 108
HPO_4_^2−^	Na_2_HPO_4_	2000	5.02	100	2000 ^a^	5.17	103
Hg(II)	Hg(NO_3_)	10	5.23	105	100	5.14	103
I^−^	KI	1000	4.84	96.9	10	4.30	86.1
Mg(II)	MgCl_2_	1000	4.73	94.7	2000 ^a^	5.06	101
Mn(II)	MnSO_4_·H_2_O	100 500	5.04 5.32	101 106	500	5.23	105
Mo(VI)	(NH_4_)_6_Mo_7_O_24_·4H_2_O	250	4.88	97.7	10	4.74	94.9
NO_3_^−^	NH_4_NO_3_	2000 ^a^	5.10	102	1000	4.71	94.4
Ni(II)	NiSO_4_·6H_2_O	0.5	5.15	103	0.5	5.10	102
Pb(II)	Pb(NO_3_)_2_	10	4.81	96.3	20	4.99	99.8
Re(VII)	NH_4_ReO_4_	250 500	4.97 4.71	99.4 94.1	5 100	4.97 4.24	99.4 84.7
Tartrate^2−^	KNaC_4_H_4_O_6_	100	4.63	92.6	100	3.63	72.6
V(IV)	VOSO_4_·5H_2_O	1	7.29	146	1	7.10	142
W(VI)	Na_2_WO_4_·2H_2_O	1	4.65	92.9	1	4.91	98.3
Zn(II)	ZnSO_4_·7H_2_O	500	5.06	101	2000 ^a^	4.83	96.7

^a^ Higher FI:V(V) mass ratio not studied. ^b^ Masked by 10 mg F^−^.

**Table 6 molecules-28-06723-t006:** Determination ^a^ of vanadium(V) in spent silica-supported catalysts.

Catalyst	Vanadium Found, %		RSD, %	
	HTAR—TTC Method	Alternative Method ^b^	HTAR—TTC Method	Alternative Method ^b^
Sample 1	2.96 ± 0.07	2.89 ± 0.12	2.4	4.2
Sample 2	2.42 ± 0.06	2.48 ± 0.11	2.5	4.4
Sample 3	2.99 ± 0.08	3.06 ± 0.11	2.7	3.6
Sample 4	1.95 ± 0.04	2.03 ± 0.09	2.1	4.4

^a^ Four replicate analyses (mean ± standard deviation). ^b^ Ref. [53].

**Table 7 molecules-28-06723-t007:** Determination ^a^ of vanadium(V) in pharmaceutical samples.

Sample	V(V) Spike, ng mL^−1^	V(V) Found, ng mL^−1^	RSD, %	*R*, %
Sterimar nasal spray	0	Not detected	–	–
	25	24.3 ± 1.1	4.5	97.2
	50	51.7 ± 1.3	2.5	103
	75	73.9 ± 1.5	2.0	98.5
Marimer inhalation	0	Not detected	–	–
	25	26.3 ± 1.2	4.6	105
	50	49.4 ± 1.8	3.6	98.8
	75	75.2 ± 2.4	3.2	100
Solution for intravenous infusion	0	Not detected	–	–
	25	25.6 ± 0.8	3.1	102
	50	50.6 ± 1.3	2.6	101
	75	74.0 ± 2.3	3.1	98.6

^a^ Three replicate analyses (mean ± standard deviation).

**Table 8 molecules-28-06723-t008:** Comparison with other spectrophotometric procedures.

Reagent(s)	Preconcentration Technique	*λ*_max_, nm	*ε* × 10^−4^, L mol^−1^ cm^−1^	Linear Range, μg mL^−1^	LOD, ng mL^−1^	Application	Ref.
APANOL	–	533	1.02	0.1–4.00	54	Rise and flour	[60]
BPHA	LLE	530	0.545	Up to 1.5	–	Water samples	[19]
CV + KI	–	588	1.27	0.1–10.2	675	Soil, biological, and pharmaceutical samples	[61]
DTP + Amines	LLE	615–620	2.8–3.0	0.05–16	–	Soils, oil, and oil products	[16]
EDTA + ST	UA-CPE	530	–	0.002–0.18	0.26	Vegetal oils and vinegar	[55]
HPMEPB	LLE	415	2.7	Up to 2.2	79.2	Synthetic and technical samples	[58]
MQ	LLE	400	0.2449	Up to 6.2	168	–	[20]
PCNPC4RAHA	LLE	495	0.554	0.137–9.36	8.89	Steels, environmental and biological samples	[21]
PG + ST	UA-CPE	533	–	0.002–0.5	0.58	Beverage samples	[17]
TA + CTAB	DLLME-SFOD	600	–	0.006–1	1.8	Fruit juice samples	[54]
TAN + H_2_O_2_	MA-CPE	607	8.84	Up to 0.76	1.4	Mineral water, pharmaceutical, and industrial samples	[53]
HTAR + NTC	LLE	556	5.2	0.023–1.1	6.8	–	This work
HTAR + TTC	LLE	549	5.2	0.015–2.0	4.6	Catalysts and pharmaceuticals	This work

Abbreviations: APANOL, 1-[(4-Antipylazo)]-2-naphthol; BPHA, *N*-benzoyl-*N*-phenylhydroxylamine; CTAB, cetyltrimethylammonium bromide; CV, crystal violet; DTP, 2,6-Dithiolphenol; DLLME-SFOD, dispersive liquid–liquid microextraction based on solidified floated organic drop; EDTA, ethylenediaminetetraacetic acid; HPMEPB, 3-Hydroxy-2-[3-(4-methoxyphenyl)-1-phenyl-4-pyrazolyl]-4-oxo-4*H*-1-benzopyran; LLE, liquid–liquid extraction; MA-CPE, microwave-assisted cloud point extraction; MQ, 2-Methyl-8-quinolinol; PCNPC4RAHA, p-Carboxy-*N*-phenyl-calix [4]resorcinarene-hydroxamic acid; PG, pyrogallol; ST, safranin T; TA, tannic acid; TAN, 1-(2-thiazolylazo)-2-naphthol; UA-CPE, ultrasound-assisted cloud point extraction.

## Data Availability

Not applicable.

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
