# Peer review of "Extractive Spectrophotometric Determination and Theoretical Investigations of Two New Vanadium(V) Complexes"

_molecules, 2023, doi:10.3390/molecules28186723_

Round 1
Reviewer 1 Report
The paper presents a study on the detection of vanadium complexes using different cationic tetrazolium dyes and attempts to establish a method for quantification.
Unfortunately, the manuscript needs extensive editing and a near-complete rewrite.
There are serious concerns with the paper, a fraction of which I present here:
Issues:
The abstract is difficult to read as it is mostly just short sentences and needs serious editing and rewrites. It is clear the authors spent a lot of time working on their project, so use the abstract to hook readers into wanting to read their paper. Rework the introduction to tell a nice summary and hook the readers attention.
For example, lines 15 – 17 can be easily rewritten to something along the lines of:
The two dye complexes exhibit distinctive metal complexes wherein the TTC forms a dimer (give formula) while the NTC complex forms a monomer (give formula).
Vague wording: In line 19 and 20 the authors state “…is characterized by some advantages…” Like what? Be specific. What did you find in your studies? Why is it advantageous?
Line 21 : “The developed procedure…” What procedure? I think the authors mean to say something along the lines of “The extraction procedure presented herein presents a cheap and reliable method for extraction and quantification of vanadium ions in solution.” Or something along those lines.
The whole first paragraph (lines 30 to 35) is pointless. You are writing to an audience of chemists. This is wholly unnecessary. Remove this. All lines through 38 can be easily removed in fact. The problem which the authors are hoping to address revolves around vanadium as a pollutant. So the introduction should summarize that, not what vanadium is.
Vague wording in line 42: “…vanadium probably ranks first among the trace elements…” Is it or is it not the first-ranked element? Later in line 46 they say its on the same level as other metals anyway.
Line 62 – The figure is incorrectly captioned. TTC and NTC are both labeled as (a)
Lines 69 – 73: The authors state in lines 69 and 70 that “…the ability of their cations depends on various factors such as molecular weight, number of tetrazolium rings…” Then in line 73 they state that “The reasons for this individuality are not always clear…” I don’t understand the point the authors are trying to make. The different dyes all have clearly different geometries and electronic structure. Of course they will behave differently, if based on nothing more than mere solubility due to distinctive polar and non-polar regions. This whole part of the discussion needs to be clarified or removed.
Line 78 – 79 makes no sense. The authors state that the maxima are “practically the same” then in the next line show they are different values. The values are not the same nor practically the same. Clarify this statement.
All of the figure captions need to be trimmed down. Move the experimental portion to the appropriate part of the paper. The captions should be simple and easy to read, not bogged down with experimental details which don’t help in the interpretation of the image. What is the image and what one or two important details does it show?
Section 2.1 presents no Results or Discussion despite being in the results and discussion section. It merely shows the graph and then ends after 3 lines.
Line 106: “About 2 min are needed…” Is it 2 minutes? Or is it 2 minutes and 5 seconds? Give exact values. I don’t understand why the authors are being vague. This is a scientific paper written for people who have a keen interest in this field.
The authors then recommend times that are different than the equilibrium times (line 109). Why?
Figures 7 and 8 are far too cluttered. You can barely see the data and the fitting line as most of the graph is taken up by text. Move the current graphs to the SI and make clean graphs with fitting for the text so people can analyze the fitting you used themselves.
Figure 11a. Where is the chloride ion? Did the authors optimize just the cation (that’s fine), but then for 11b there is just one chloride ion when there should be two? This needs to be clarified.
Why did the authors use different levels of theory for the optimizations of the cations?
Why switch to HF for the dimer study? If it’s a computational limitation, then that should be stated and addressed. However, in the acknowledgments, they state that an HPC was used for the calculations. I don’t see why higher-level theories weren’t used then. Particularly when the authors use non-covalent interactions as part of their discussions (stacking interactions). B3LYP is not the best choice for methods in this case.
Moreover, did the authors do the optimizations in solution or gas-phase? This is not clarified anywhere that I could see.
Extensive editing will need to be done prior to the next submission. While the manuscript is certainly readable, there needs to be heavy clarification and improvement on nearly every part of the text.
Reviewer 2 Report
In this work, Two new complexes of vanadium(V) involving an azo dye (6-hexyl-4-(2-thiazolylazo)res orcinol, HTAR) and a tetrazolium cation were studied. The cations were derived from the following commercially available salts: Tetrazolium Red (2,3,5-triphenyltetrazol-2-ium;chloride, TTC) and Neotetrazolium chloride (2-[4-[4-(3,5-diphenyltetrazol-2-ium-2-yl)phenyl]phenyl]-3,5-diphenylte- trazol-2-ium;dichloride, NTC). These cations impart high hydrophobicity to the ternary complexes, allowing vanadium to be easily extracted and preconcentrated in one step. The chloroform-ex- tracted species differ in stoichiometry and properties. The complex of Tetrazolium Red dimerizes in the organic phase. It can be represented by the formula [(TT+ )[VO2(HTAR)]]2. The other complex 16 is a monomer: (NTC+ )[VO2(HTAR)].
The manuscript must be well revised based on the following comments to be reconsidered in case of acceptance:
1. The whole manuscript still needs to be double checked in terms of language issues (scientifically and grammatically).
2. Title should be extended in terms of what type of factors you exactly explored. It seems pretty general.
3. The beginning of abstract should also extend and highlight the major concern as a whole view to the reader.
4. Too many keywords! No more than 5 is suggested.
5. The first sentence of the introduction: how essential? Any scientific data? Without reference? Please talk scientifically about such claims, even if it is very well-known. Check the whole manuscript in terms of such things.
6. The last paragraph of the introduction should focus on the remarkable points of the current work within a brief introduction to the whole project, highlighting its novelty.
7. The environmental concerns referred to dye compounds should be also addressed in the introduction. Some references could be taken into consideration like:
Journal of Alloys and Compounds 852 (2021) 156955, https://doi.org/10.1007/s10904-023-02693-x
8. Significant figures should be appropriately considered in reporting all quantitative data (tables, in particular).
9. Table 6, caption: in spent silica…, spent! What do you mean? Please check all captions, similarly.
10. Table 7, recovery % of higher than 100%! Is that meaningful?
11. Subsection 3.8 is just an adjective.
12. Please compare the first sentences of the conclusion and abstract to well judge of addressing your concerns to perform this work. In your conclusions, please discuss the implications of your research and make sure your conclusions' section underscores the scientific value-added of your paper. Please add some scientific key values of your results to the conclusion. What would be the impact of your investigation in the real world? Add a sentence into the conclusion please.
Good luck
needs revision
Reviewer 3 Report
The manuscript by Kiril B. Gavazov and co-workers entitled “Experimental and Theoretical Investigations of Two New Vanadium(V) Complexes” describes the work done with two tetrazolium salts (TCs) and an azo die, which are used to extract V(V) from aqueous solutions with chloroform. The manuscript is very interesting and the works seems to be well carried. The manuscript is quite long and has too many figures. Some figures could be sent to supplemental materials.
However, I must inform that I am not an expert in theoretical calculations and thus cannot evaluate the theoretical studies.
My 1st question is: should chloroform be used in such an application given its volatility, toxicity and carcinogenity?
The title should be more informative as it’s too general and does not reflect the work done.
45 – “…as the most common anthropogenic vanadium form – V(V) – is a toxicant on the same level as mercury, arsenic, lead” I am a bit surprised by this statement. Toxicity depends on a lot of factors, and this seems like an overstatement. Please rephrase it.
71 – “TSs show individuality..” Is individuality the right word?
Figure 2 – I do not understand what are the blanks and what does this figure really shows. Please explain in the text what’s the aim of this experiment.
90- Does ammonium acetate have buffering capacity in all this pH range?
134 – Figure 2 not 1
192, 193 – Please replace “=” signals by arrows. What is the meaning of (o)?
284 – In general there is specificity for V(V) but when in the presence of V(IV) 1:1 the % of recovery is ca. 145. How do you explain this?
Section 3.2 does not contain instrumentation but repeats section 3.1
The language could be improved
Round 2
Reviewer 1 Report
The authors appear to have addressed most major concerns presented in the previous reviews.
Author Response
Thank you!
Reviewer 2 Report
Authors have well addressed the comments. Accepted.
minor edition is needed
Author Response
Thank you!